# Thermal Comfort and Electrostatic Properties of Socks Containing Fibers with Bio-Ceramic, Silver and Carbon Additives

**DOI:** 10.3390/ma15082908

**Published:** 2022-04-15

**Authors:** Laimutė Stygienė, Sigitas Krauledas, Aušra Abraitienė, Sandra Varnaitė-Žuravliova, Kristina Dubinskaitė

**Affiliations:** Department of Textile Technologies, Center for Physical Sciences and Technology, Demokratų Str. 53, LT-48485 Kaunas, Lithuania; laimute.stygiene@ftmc.lt (L.S.); sigitas.krauledas@ftmc.lt (S.K.); ti@ftmc.lt (A.A.); kristina.dubinskaite@ftmc.lt (K.D.)

**Keywords:** thermal effectiveness, thermoregulation, bio-ceramic, far infrared, silver, carbon, electrostatic, socks

## Abstract

Socks are an important part of our clothing used in everyday activities. In order to ensure thermal comfort during wear in cool outdoor or indoor conditions, and for health improvement, socks must have effective thermoregulation properties. Chemical far-infrared (FIR) fibers with different bio-ceramic compounds incorporated into socks’ structures can provide an improved thermoregulation effect to the wearer of the socks. Fibers with silver and carbon additives incorporated in their structures can also affect the thermoregulation properties of socks. Moreover, these conductive additives avoid the unpleasant effect of static electricity of socks. The main parts of the different investigated structures of the socks were made in a plush pattern. The plush loops were formed by using functional Resistex^®^ Bioceramic, Shieldex^®^ and two modifications of Nega-Stat^®^ fiber yarns. The main thermal comfort (thermal efficiency, microclimate and heat exchange temperatures, thermal resistance, water vapor permeability) and electrostatic (surface and vertical resistances, shielding factor, half time decay of charge) properties of the socks were investigated. Based on the obtained results of the thermal comfort and electrostatic characteristics of the different investigated structures of socks, the optimal static dissipative (half-time decay <0.01 s, shielding factor—0.96) plush knitting structure with 55% Resistex^®^ Bioceramic and 31% bicomponent Nega-Stat^®^ P210 fibers yarns was selected. Comparing the control sample without FIR and the knitted structure with conductive additives, we can draw the conclusion that the heat retention capability of the selected socks was improved by 1.5 °C and the temperature of their created microclimate was improved by 2 °C.

## 1. Introduction

Socks are an important part of our clothing, and are necessary for everyday activities in order to avoid the discomfort of the foot due to moisture coupling in the shoes during continuous wearing [1,2]. It is necessary to consider aspects of thermoregulative effectiveness while designing them, because the wearing of socks allows less air circulation in shoes than in other areas of the body. The most important factors which influence the thermophysiological comfort properties of textile materials are the fiber content of the raw materials used, the geometrical properties and structure of the yarns used, the structure and parameters of the fabric, and the finishing character [3,4,5]. In order to maintain a constant physiological body temperature, the emitted sweat must be transferred to the outside of the garment (including socks) at the rate at which heat is released by physical activity [6]. Many scientific papers analyze the dependence of the thermophysiological properties of single-layer and double-layer knitted fabrics on their fiber composition, the structure and its parameters [7,8,9,10,11,12,13,14]. Studies of the fiber composition, pattern and structural parameters of bilayered weft knitted fabrics on thermophysiological comfort [15,16,17,18] have shown that the water vapor permeability is mainly influenced by the fiber composition, by up to 14%, while the knitting structure only has up to an 8% influence. As thermal comfort is a very important parameter for human health, a lot of research [19,20,21,22,23] has also recently been devoted to the investigation of the thermoregulatory properties of socks. It should be noted that, in terms of thermal comfort, the same regularities are valid for socks and for knitted fabric. The increased warming effect of socks is important for consumers during wearing in cool outdoor or indoor conditions, for recreational activities, or for the improvement of the health of the non-active elderly or disabled people, as well as for the warm-up phase of the athlete by increasing muscle efficiency and reducing the risk of injury [24,25,26,27,28]. Bio-ceramic additives embedded in the structure of the socks can be used in order to improve their thermal properties. Textile materials with these additives are able to absorb and emit back the thermal energy from human skin. Far-infrared (FIR) textiles are a new category of functional textiles that have presumptive health and wellbeing functionality [24]. According to literature data, FIR radiation (3–100 μm) has biological activity, and in the narrow range of 3–12 μm it is also applied for therapeutic purposes [29]. It has been found that only in the range of 8–12 µm can FIR energy be absorbed and delivered to the human body in the form of heat [30]. The most important part of IR radiation is the far-infrared spectrum of 6–15 μm, which is responsible for creating the human sense of heat. This part of the IR radiation corresponds to the radiation of the human body itself. For this reason, any external radiation of such a wavelength is accepted by the body as “its own”.

High thermal efficiency materials are obtained by the usage of functional fibers with embedded far-infrared (FIR) bioceramic additives with heat storage/release capabilities [31], or by the incorporation of these additives into the structure of the material with surface chemical modification processes [32]. Clothing with improved thermoregulatory properties is widely used in health, medicine, and warm-up sports [33,34,35,36,37]. Magnesium, zirconium and iron oxides, silicon carbide and germanium compounds, or other minerals, acting on the principle of microspheres and nanoparticles, are introduced into fabrics that are primarily worn next to the skin (to reflect FIRs back to the body) in order to increase thermal effectiveness [38,39]. The electrostatic properties of socks are important in terms of their antistatic aspect [23]. As is already known, bio-ceramic additives can be incorporated during spinning by the formation of chemical fibers, which have a tendency to accumulate static electricity [40]. Knitted or woven fabrics with the above properties can be produced using a variety of functional fibers with incorporated silver, copper, steel or carbon additives. These conductive additives can absorb and disperse the uncomfortable electric charges that accumulate from the external environment. The antistatic properties may also be conferred by special finishing by treating the materials with antistatic preparations with anionic and cationic agents with a long hydrocarbon chain with ionic groups, e.g., carbon, carbon nanotubes, copper, polypyrrole, or polyaniline [41].

It should be noted that fibers with embedded electrically conductive additives not only prevent the accumulation of static electricity in textile materials but also absorb and release infrared rays, and due to their provision of increased thermal effect they improve the thermal comfort properties of textiles [42,43,44]. For example, the IR reflection factor of silver is more than 95%. Therefore, silver additives incorporated into textiles minimize heat loss by reflecting their own energy from the skin during wearing. Silver also has a very low thermal energy radiation rate. The presence of silver additives in clothing means that the heat reflected from the body is absorbed and stored in the fiber, resulting in a warm feeling for the wearer for a longer period [43]. It is also stated [44] that, typically, conductive textile materials transfer heat flow faster than non-conductive materials. This means that conductive additives such as metals can increase thermal conductivity, decreasing the thermal resistance simultaneously. This explains the claim that silver-containing textile materials are warm in winter and cool in summer. According to the scientific literature [33,45], another class of FIR fibers are bamboo-carbon modified polyesters (BCMP). Due to the presence of incorporated bamboo-carbon powders, BCMPs are reported to have good FIR emissivity properties. It is claimed that the use of BCMP fabrics can raise the body temperature by 4–7 °C.

The goal of this research is to investigate the thermal comfort and electrostatic properties of designed socks containing different combinations of fibers with bioceramic, silver and carbon additives, and to select the optimal knitted structure of the socks in order to provide greater thermal comfort and antistatic properties for the wearer. The novelty of the research is that upon performing a scientific data search regarding functional knitted fabrics, produced by using in their plush structure combinations of functional FIR, carbon and silver fibers, no papers about analogous investigations—especially socks—were found.

## 2. Materials and Methods

Taking into account that the research is based on the improvement of the functional thermoregulation and electrostatic properties of socks for their design and development, the following raw materials were selected:19.7 tex polyester (PES) spun yarns, Resistex^®^ Bioceramic (Tecnofilati S.r.l., Bergamo, Italy);7.8 tex (filament count—12) bicomponent PES/C, multifilament yarns with a trilobal-shaped carbon core, Nega-Stat^®^ P190 (W. Barnet GmbH & Co. KG, Aachen, Germany);11.2 tex (filament count—12) bicomponent PES/C multifilament yarns with a trilobal carbon cross section, which 3 segment exits to the PES fiber surface, Nega-Stat^®^ P210 (W. Barnet GmbH & Co. KG, Aachen, Germany);17.0 tex twisted (filament count: 32 + 12) PES silver (Ag)-coated multifilament yarns, Shieldex^®^ (Statex Productions & Vertriebs GmbH, Bremen, Germany);11.1 tex textured PES yarns (Hangzhou Tita Industry Co., Ltd., Zhejiang, China);7.8 tex PA + 2.2 tex elastane (EL) air-intermingled yarns (Mistral—Elast Sp. z o.o., Łódź, Poland);

The selected PES spun yarns—Resistex^®^ Bioceramic with fiber-embedded bioceramic (titanium, zinc, zirconium, silicon, calcium, etc.) infrared-emitting particles [31]—enhance the wearer’s thermal comfort by providing an additional thermal effect.

Shieldex^®^ twisted PES yarns consist of two components—polyester 11.3 tex (filament count—32) and polyester 4.4 tex (filament count—12). The latter 12 filaments were coated with a very thin layer of silver (Ag). These yarns were used in knits, giving them permanent antistatic and thermoregulation (IR reflection, heat retaining, acceleration of moisture evaporation) properties.

The selected Nega-Stat^®^ core- and surface-conducting PES bicomponent yarns with different forms of carbon-core for each filament, used in the knits, can also provide static-dissipative and improved thermal properties [46].

In order to increase the thermal efficiency and to provide the antistatic properties of socks, the newly designed yarns were produced using a PL-31 ring-twisting machine. The main characteristics of the prepared two-ply twisted yarns are presented in Table 1.

According to the method of manufacturing the socks, they are classified as finished (regular) knitwear [47], The main parts of the socks are as follows: (1) cuff, (2) leg, (3) heel, (4) sole, (5) toe, and (6) foot; the spreading scheme of the yarns is as follows: (7) ground yarn, and (8) plush yarn; these are presented in Figure 1.

Socks of 5 structures were manufactured on the single-cylinder sock knitting machine Fantasia (Sangiacomo S.p.A., Brescia, Italy) of gauge 14E (diameter of the cylinder, 3.75″; number of needles, 168). Good thermoregulatory and antistatic properties are important in both the foot and the calf area of the socks. The parts of the socks with the plush fabric structure and marked as 2–6 in Figure 1 were examined during the research, because they give the best thermoisolation properties primarily. The cuff of the socks was not examined, because it keeps the sock from rolling down or slipping from the leg, is usually elastic, and has a rib/double-layer knitted structure with incorporated elastane. Besides this, by forming plush loops into the knitted structure, a relatively large quantity of fibers with ceramic, silver and carbon additives can be inserted. The schematic view of this pattern is shown in Figure 1. The structure parameters of the socks manufactured are presented in Table 2. It is taken that the plush surface will be in contact with human skin during wear. The control sample No. 5 was produced from regular PES yarns in plush loops for the comparison of the thermal comfort characteristics with the results of other socks containing functional additives.

All of the fabricated socks (see Table 2) were washed with 2.5 g/L non-ionic detergent Felosan RG-N, 2.0 g/L sodium carbonate (Na_2_CO_3_), 3.0 g/L water softener Calgon^®^ Power at 60 °C for 60 min in a washing machine (WASCATOR FOM71MP), rinsed with 1 g/L acetic acid water solution, centrifuged, and air flat dried.

In this work, the following thermoregulatory properties of ceramic, silver and carbon additives containing knitted structures of socks were investigated: heat storage/release capability, microclimate and heat exchange temperatures, thermal resistance, and water vapor permeability. Furthermore, the parameters of the electrostatic properties—such as surface and vertical resistances, shielding factor, and half decay time—were evaluated.

Determination of the parameters of the knitted sock’s structure: The mass per unit area of the socks was determined according to EN 12127, and the number of stitches per unit length and unit area were determined according to EN 14971 standards.

Evaluation of the heat storage/release capability: For the evaluation of the thermal effect of socks with bioceramic, carbon and silver additives, acting on the principle of FIR radiation, the original method for the estimation of the heat accumulation/release capacity under a changing ambient temperature was used (see Figure 2). The sample was placed on an expanded polystyrene (EPS) plate so that the inner plush surface of the sock faces an intense heat source (IR lamp: power 250 W, diameter Ø = 25 cm; distance between sample and heat source—80 cm, ambient temperature 22 ± 0.5 °C). Simultaneously, the heat flow outgoing from outside of the test sample was isolated with an EPS plate. The Infra Cam^TM^ (FLIR Systems AB, Täby, Sweden; EM spectral range 7.5–13 µm, accuracy 2% of reading, thermal sensitivity 0.2 °C, emission factor 0.95) was used to capture a thermal image of the sample when the lamp was switched on, at intervals of 15 s and 4 min (240 s). When the lamp was off and the specimen cooled down (every 15 s for 4 min (240 s)) the thermal images were captured as well. Flir Reporter 9.0 software was used for the processing of the received data. The mean apparent temperature of the sample after 5 measurements of the hottest area was calculated for each sample. The coefficients of variation of the results of the temperature measurements did not exceed 5%.

Assessment of the Thermal Efficiency on the hot plate simulating the temperature of the human skin: The temperature of human skin depends on the environmental conditions, and varies in different parts of the body from 32 °C to 37 °C; the average human skin temperature is assumed to be about 34 °C [48,49]. Therefore, investigations of the microclimate and heat exchange temperature parameters of the developed samples were carried out (room temperature—22.0 ± 0.5 °C, relative humidity—35 ± 1%) on the hot plate, heated up to 34 °C, at which the temperature of the human skin was simulated. The test samples were placed on the hot plate such that the plush surface of the inside of the socks was in contact with the plate. In this way, we simulated the skin-surface/sock interaction during wear. The warming temperature parameters were recorded over time at intervals of 10 s for 5 min (300 s) using a thermocouple-type sensor 2 × NTC Type N (measuring range −50 to +125 °C) which was connected with the help of a connector (ZA 9040 FS) to the ALMEMO^®^ 2470 series universal measuring instrument (Ahlborn Mess- und Regelungstechnik GmbH, Holzkirchen, Germany) with 2 measurement inputs and a long-term data logger. The first (external), thermocouple sensor 1, was fastened on the outside of the surface of the socks (see Figure 3). The temperature recorded by this sensor characterizes the heat exchange through the fabric. The second (internal), thermocouple sensor 2, was superimposed between the hot plate and the plush surface of the knitted socks, and recorded the created microclimate temperature, which refers to the air layer nearest to the skin. The mean values out of 5 measurements were given as a result. The calculated coefficient of variation was less than 5%.

Evaluation of the thermal resistance and water vapor permeability: The measurements of the thermal resistance of the knitted socks were performed in a standard atmosphere: temperature (20 ± 2) °C, relative humidity (65 ± 4) %. The thermal resistance was determined by a standardized procedure, EN ISO 11092, using a Sweating Guarded Hot Plate M259B device (SDL International Ltd., Stockport, UK), simulating human skin, of which the basic part was a porous metal plate which heats up to 35 °C, i.e., the human body temperature (see Figure 4). The metal plate is approximately 3 mm thick, and has an area of 0.04 m^2^. The test was carried out on three specimens, as required according to EN ISO 11092 standard. The coefficients of variation of the results of the thermal resistance did not exceed 5%.

For the determination of the water vapor permeability, a modified “cup” method (Figure 5) was used [50]. The measure of the characteristic is the amount of water evaporated from the covered vessel in 24 h. The equipment used was a thermostat U10 with a water bath maintained at 38 °C, a thermostatic receptacle made of steel with an inside diameter of 88.2 mm, a 500 mL measuring container, and weighing scales with an accuracy of 0.01 g. The specimens were prepared as follows: three specimens of 115 mm diameter were cut and weighed from different places of the material maintained in a standard climate for 24 h. Then, a dish with 500 mL distilled water was placed in the thermostat. The sample was firmly secured to the pan for 6 h. The ambient temperature during the tests was—22.0 ± 0.1 °C. The coefficients of variation of the studied water vapor permeability did not exceed 8%.

Determination of the electrostatic properties: The parameters of electrostatic properties such as the surface and vertical resistance, shielding factor and half decay time were measured. The surface resistance and specific surface resistivity of the knitted fabrics were determined according to the EN 1149-1 standard using a suitable device (Terra-Ohm-Meter 6206 (Eltex-Elektrostatik-GmbH, Weil am Rhein, Germany)) with a measuring range of 10^3^–10^14^ Ω. The materials were tested in a chamber at 23 °C with a relative humidity of 25% and a reading time of 3 s for the test value. The vertical electrical resistance of the knitted fabrics was determined according to the EN 1149-2 standard using the same device—a Terra-Ohm-Meter 6206. The concentric ring probe was used for the measurement of the electrical resistances, which provided a result in Ohms (Ω). The mean values out of five specimens were determined. The coefficient of variation of the resistances’ values did not exceed 5.0%. The measurements of the shielding factor and half decay time of the test materials were performed according to the requirements of EN 1149-3, second method (induction charging), using an apparatus for the determination of the charge decay of textiles (ICM-1 (STFI, Chemnitz, Germany)) with a Super Dry SD-151-02 climate camera at 23 °C temperature and a relative humidity of 26–27%. Samples of the socks were conditioned for 24 h before the experiments in a humidity and temperature-controlled chamber (JCI 191), with an air temperature of 23 ± 1°C and a specific humidity of 25 ± 5%. The instrument ICM-1 was controlled by a microprocessor which made measurements, provided automatic calculations, and displayed the measured data. The coefficients of variation of the measured parameters of each of the 5 samples were not more than 1.5%. The values of the shielding factor were calculated according to the values of the maximum electric field strength, as indicated by the recording device with the test specimen, and of the electric field strength, as indicated by the recording device without test specimen in the measuring position.

## 3. Results

It is well known that the increased thermal efficiency of textile materials is related to the presence of functional additives such as ceramic, carbon, silver. The additional thermal effect can be created due to the ability of these additives to reflect the thermal energy generated by the IR rays. Thermal efficiency evaluation data of the manufactured socks, obtained by an IR light-emitting lamp and a thermal imaging camera, are shown in Figure 6, Figure 7 and Figure 8. This test simulates the stored amount of heat which can be reflected back to the human body from the inner surface of socks. The analogous methodology to determine the heat storage/retention capability generated by IR rays of textile materials was used by the authors in [26,31]. These studies showed that bioceramic additives incorporated in the structure of textile materials can provide higher thermal efficiency. The highest IR ray absorbing and retaining values (warming temperature 53.6 °C) were determined in a fabric in which all three layers were composed from Resistex Bioceramic yarns [31].

The graphs (see Figure 6 and Figure 7) and thermal images of the heating process (Figure 8) showed that the highest heat accumulation was determined after 240 s heating for structures No. 2 (warm-up temperature 39.2 °C) and No. 3 (warm-up temperature 37.6 °C). Control sample No. 5, manufactured from regular polyester yarns, had a maximum warm-up temperature of only 29.3 °C, as it had no additives which could capture heat. The additional thermal effect was provided due to the presence of an incorporated 55 ÷ 62% Resistex^®^ Bioceramic and two modifications of 31 ÷ 25% Nega-Stat^®^ fiber yarns. Among the conductive knitted structures, the smallest additional thermal effect compared to control sample No. 5 and sample No. 4 was achieved in sample No. 1, containing 48% Resistex^®^ Bioceramic and 42% Shieldex^®^ fibers (maximum warm-up temperatures 29.3 °C, 29.9 °C and 36.6 °C respectively). Comparing the maximum warming temperatures (29.9 °C and 29.3 °C, respectively) of sample No. 4 (with bioceramic additives) with the control sample No. 5 (without bioceramic additives), it can be stated that the presence of 91% Resistex^®^ Bioceramic fibers in the structure of the socks leads to a very small additional (+2.05%) thermal effect. Much poorer results (almost twice) were obtained in a research article [24]. The authors determined that the heat accumulation rate of knitted samples containing bioceramic additives inside the fiber was slightly higher (+0.6%) compared to the control sample without incorporated bioceramic. The results obtained testify to the ability of the designed socks to provide an additional thermal effect due to the presence of incorporated Resistex^®^ Bioceramic, Nega-Stat^®^ and Shieldex^®^ functional yarns. However, the greatest additional thermal effect (+(33.79 ÷ 24.91)%) was provided by conductive Nega-Stat^®^ and Shieldex^®^ fibers. The carbon and silver particles captured and released heat much more effectively than the bioceramics.

In order to evaluate the thermal efficiency properties of socks, the preservation of accumulated heat release during the cooling process is also a very important parameter. The temperatures recorded during the cooling process of socks are shown in Figure 6 and Figure 7. It can be seen that at the end of the cooling process (t = 480 s) the recorded surface temperatures of samples containing fibers with ceramic and conductive additives are in the range of 24.0 °C to 24.8 °C (i.e., the average percentage decrease of temperature’s rate is +3.00% to +6.44% higher compared to control sample 5). Summarizing the investigations of the thermal efficiency in the heating–cooling phases, it can be seen that samples No. 2 and No. 3 (in composition containing 62% ÷ 55% Resistex^®^ Bioceramic and 25% ÷ 31% Nega-Stat^®^ fibers) have the most remarkable thermal efficiency. The improvement of +4.26% in the heat accumulation of sample No. 2, with 25% Nega-Stat^®^ P190 core conducting with trilobal-shaped carbon core fibers, compared to sample No. 3—consisting of 31% Nega-Stat^®^ P210 surface conducting fibers—was also determined. However, the graphs (Figure 6 and Figure 7) show that after irradiation, sample No. 3 has slightly better heat retention properties compared to sample No. 2 (the recorded temperatures at the end of cooling process were respectively 24.8 °C and 24.4 °C).

The assessment of the thermal efficiency of the socks in other environmental conditions according to the methodology presented earlier—on the hot plate simulating temperature of the human skin, with a temperature of 34 °C—was also performed. This method better simulated the real wearing conditions. A similar methodology to determine the heat exchange was used by authors in [51]. The authors investigated different multi-layered structures of wool/PES knitted fabrics, and received good thermal insulation results: none reached 40 °C—the temperature of the heating plate—after a 1 h period. The results measured by the thermocouples’ microclimate (between the plate and the inner knitted surface) and heat exchange (on the outer side of the knitted surface) temperatures are presented in Figure 9, Figure 10 and Figure 11.

It can be seen from the results presented in Figure 10 that the steady state microclimate temperatures were reached after approximately 150 s. The best results of microclimate temperatures after 300 s were shown samples No. 3 and No. 2 (temperatures of, respectively, 35.3 °C and 34.44 °C). This effect can be explained by the presence of Resistex^®^ Bioceramic (samples No. 3, 2), Nega-Stat^®^ P210 (sample No. 3) and Nega-Stat^®^ P190 (sample No. 2). Samples No. 4 (containing 91% Resistex^®^ Bioceramic) and No. 1 (containing 48% Resistex^®^ Bioceramic + 42% Shieldex^®^) showed slightly worse results. However, the steady state microclimate temperatures created for all of the samples with functional additives were 0.43 ÷ 2.1 °C (+ (1.29 ÷ 6.3)%) higher compared to control sample No. 5 (microclimate temperature—33.32 °C).

The results of the measured heat exchange through knitted samples are presented in Figure 11. Primarily, it can be seen from the results that all of the investigated samples showed good thermal isolation results, because after 300 s none of the samples reached the temperature of the hot plate (34 °C). The outer surface of control sample No. 5 reached the highest temperature (31.16 °C), and the lowest (29.33 °C) was reached by sample No. 3. As expected, the linear dependence between microclimate and heat exchange temperatures was determined (Figure 11). The dependence shows a very strong negative correlation (coefficient of determination (R^2^)—0.9173; correlation coefficient—0.9578).

The data of the measurements of thermal resistance (R_ct_) determined for all of the samples according to the standard EN ISO 11092 are presented in Figure 12. The values of the thermal resistance of the socks varied from 0.065 to 0.089 m^2^K/W. Therefore, according to the standard CEN/TR 16422 regulation for knitwear which comes into direct contact with human skin, all of the developed samples were suitable to wear in cold climate conditions (because R_ct_ ≥ 0.05 m^2^K/W). Furthermore, from Figure 14 it can be seen that the best thermal insulation capacity was possessed by samples No. 3 and No. 2. These samples have Resistex^®^ Bioceramic (respectively 55% and 62%) and 31% Nega-Stat^®^ P210 (sample No. 3) or 25% Nega-Stat^®^ P190 (sample No. 2) fibers in their structure.

Analyzing the obtained thermal efficiency results determined by different methods, a tendency was found: higher reached temperatures (of heat absorption/retention by exposure with IR lamp, and of microclimate) resulted in higher values of R_ct,_, respectively. However, the dependencies (Figure 13a,b) between thermal resistance and recorded heat absorption/retention temperatures showed different positive correlations: moderate (Figure 13a; correlation coefficient 0.6627) and strong (Figure 13b; correlation coefficient 0.7635).

Furthermore, Figure 14a,b presents the estimated linear dependencies of the created microclimate and heat exchange temperatures through knitted samples vs. the thermal resistance parameters. In both cases, these dependencies show very strong positive and negative correlations (coefficients of determination (R^2^) respectively 0.7996 and 0.7351; correlation coefficients 0.8942 and 0.8573).

The next step of this research was to determine the water vapor permeability transmitted through the investigated samples. It is known that the water vapor permeability (“breathing”) of the textiles depends on their structure (the type of raw material and yarns, knitting pattern), structural characteristics (loop length, density, porosity), environmental conditions, and the nature of the finishing [52,53,54,55,56]. As the porosity—i.e., the air spaces between the structural parts—of the material increases, the water vapor permeability rates also increase. Most authors of scientific publications report a strong correlation between these two factors. The water vapor permeability values of the newly designed socks, determined by the cup method, are given in Figure 15. Analyzing the data obtained, compared with the control sample No. 5, the water vapor permeability values of the socks are approximately 1.73–1.40 times smaller. However, analysis of the scientific literature [52,53,54,55,56,57,58] shows that the values of the water vapor permeability (1516 ÷ 2056 g/m^2^ × 24 h) provide a good “breathability” function.

The amount of moisture (sweat) released by the body is defined by the rate of water vapour permeability. It depends on the structural characteristics of the fabric as well as the structural and hydrophobic/hydrophilic properties of the used fibers. In this case, the socks with good “breathability” were made in the same plush pattern using only hydrophobic polyester (PES) fibers (moisture regain 0.4%), and can absorb only a very small amount of moisture. For this reason, no hydrophobic or hydrophilic properties were investigated.

The results of the investigated electrostatic properties according to standards EN 1149-1, -2, -3 are presented in Table 3. The results show that highly efficient electrostatic charge dissipative (antistatic) properties were possessed by the conductive samples: No. 1, containing 42% Shieldex^®^ fibers; No. 3, containing 31% Nega-Stat^®^ P210 fibers; and No. 2, containing 25% Nega-Stat^®^ P190 fibers.

The electrostatic properties of these first two samples meet the two requirements of the EN 1149-5 standard, e.g., their half-time decay is t_50_ < 4 s and the shielding factor S > 0.2 after testing by the inducting charging method described in the EN 1149-3 standard. Sample No. 2 has a surface resistivity of 1.36 × 10^6^ Ω, and is therefore also classed as an electrostatic dissipator, because it fulfils the requirement of EN 1149-5: when tested in accordance with EN 1149-1, the surface resistivity of the knitted fabric was <2.5 × 10^9^ Ω.

## 4. Conclusions

The main thermal comfort and electrostatic properties of different structures of developed socks containing fibers with FIR-emitting bioceramic, silver and carbon additives were analyzed.

The analysis of the results of the thermal efficiency properties determined by different methods showed that the dependency between the parameters obtained of all of the structures of the investigated socks can be established. Thus, a strong negative correlation between the created microclimate and heat exchange temperatures was determined. Higher reached temperature values (except the heat exchange temperature through the socks) resulted in higher values of thermal resistance. However, the strong positive correlation between heat retention and microclimate temperatures with the thermal resistance was stated. Respectively, the dependency between the heat exchange temperatures and thermal resistance of all of the investigated socks showed a very strong negative correlation.

According to the results, it can be concluded that all of the newly developed plush socks in compositions containing functional Resistex^®^ Bioceramic, Nega-Stat^®^ and Shieldex^®^ fibers, in reference to the standard CEN/TR 16422, can be used for thermal insulation in cold climate conditions, and for health improvement in indoor environments. This was achieved through the presence of bioceramics, carbon and silver additives in the yarns that retain heat by ensuring thermal comfort and improving thermal efficiency. However, the best thermal efficiency properties of static dissipative socks were determined in sample No. 3, in its structure containing 55% Resistex^®^ Bioceramic and 31% Nega-Stat^®^ P210 fibers (thermal resistance 0.089 m^2^ K/W). The heat retention capability under the exposure of the IR irradiation at the end of the cooling process of these socks compared to the control sample was +1.5 °C higher (recorded temperatures 24.8 °C and 23.3 °C respectively). The microclimate temperature reached between the skin and these socks was the highest (35.3 °C), and the heat exchange temperature on the outer surface was the lowest (29.33 °C). The structural characteristics of the socks results in good breathability (water vapor permeability 1729 g/m^2^ × 24 h), and due to the presence of conductive additives it exhibits highly efficient electrical charge dissipation (half-time decay t_50_ < 0.01 s, shielding factor S—0.96). The achieved properties of the best socks selected are mainly affected by incorporated PES Resistex^®^ Bioceramic and bicomponent PES/C Nega-Stat^®^ (with a trilobal carbon cross section, with three segment exits to the PES fiber surface) fibers.

## Figures and Tables

**Figure 1 materials-15-02908-f001:**
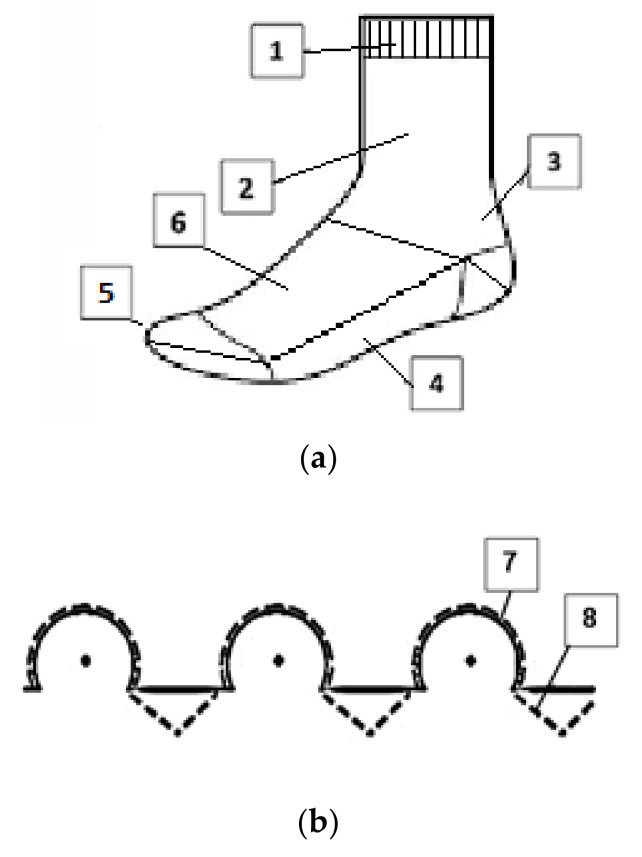
The main parts of the socks (**a**) and the spreading scheme of the of the yarns (**b**).

**Figure 2 materials-15-02908-f002:**
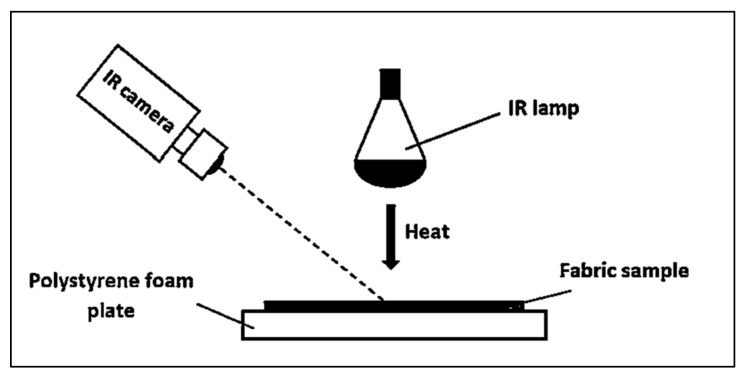
The scheme of thermal imaging equipment with IR lamp [26].

**Figure 3 materials-15-02908-f003:**
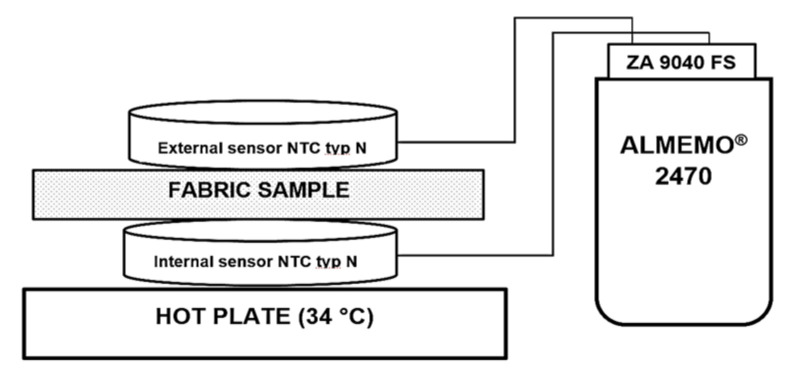
Principal measuring scheme of the external (1) and internal (2) thermocouple sensor equipment on the hot plate.

**Figure 4 materials-15-02908-f004:**
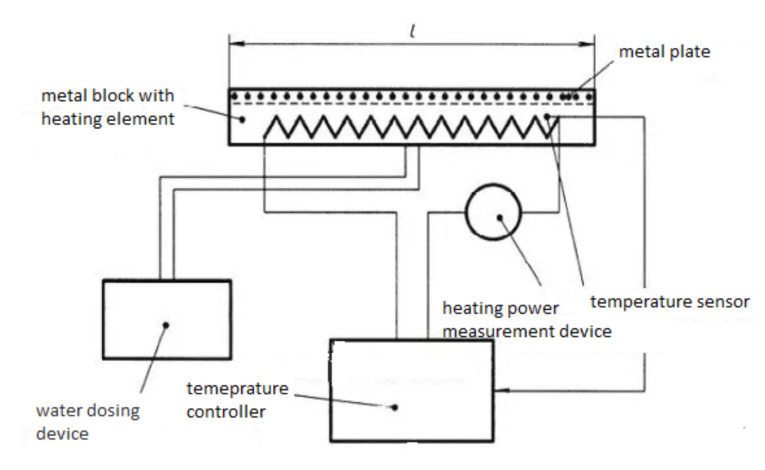
Device for the determination of the thermal resistance of fabrics.

**Figure 5 materials-15-02908-f005:**
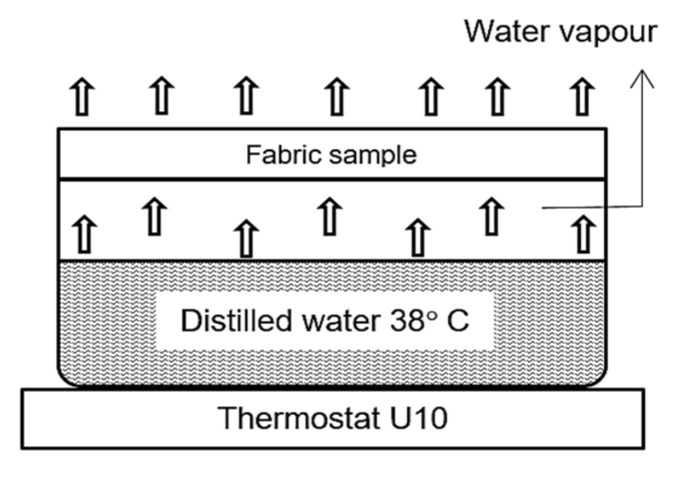
Water vapor permeability test (“cup” method).

**Figure 6 materials-15-02908-f006:**
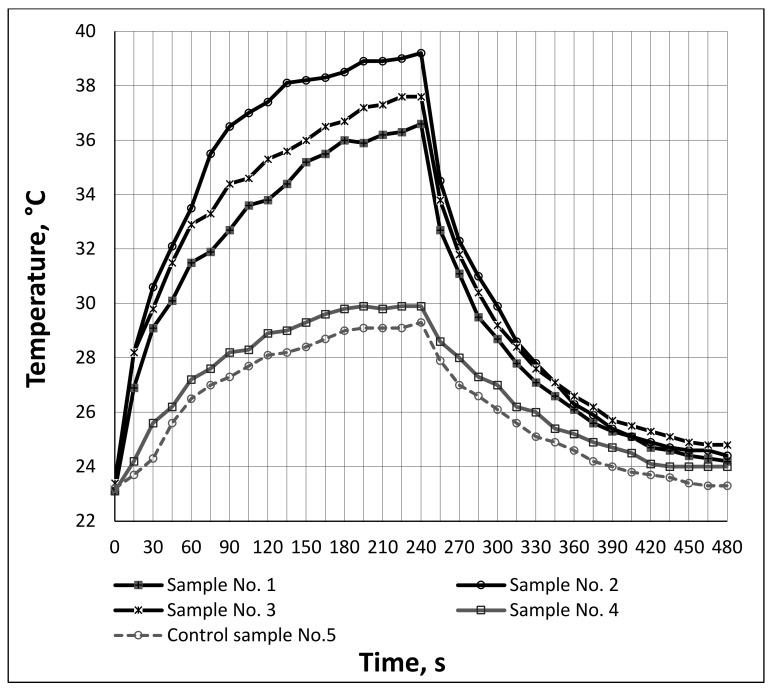
Dynamic curves of the heat accumulation/release generated by IR rays of different structures of socks.

**Figure 7 materials-15-02908-f007:**
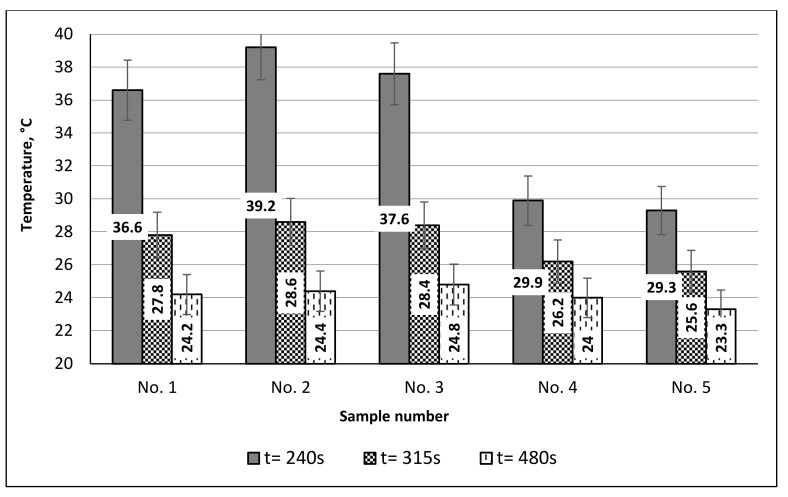
Rate of heat absorption (at time 240 s) of knitted structures of socks by exposure to a 250 W IR lamp, and heat retention (at times of 315 s and 480 s) after irradiation.

**Figure 8 materials-15-02908-f008:**
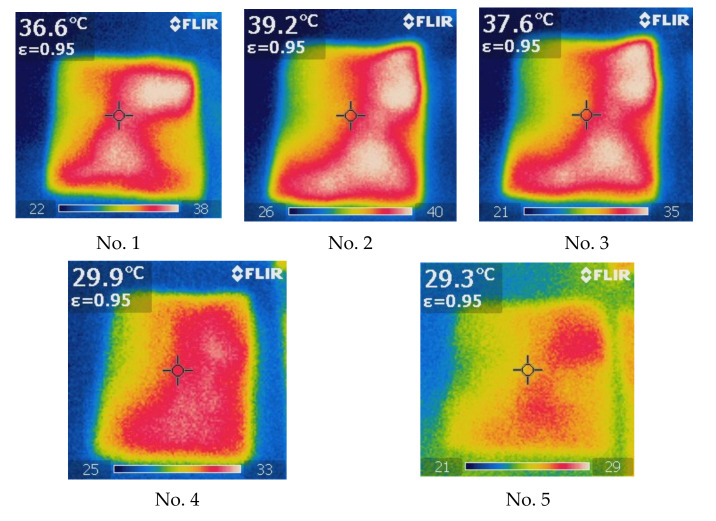
Thermographic images of the measured heating temperatures of the test samples No. 1–No. 5, when the IR exposure time t = 240 s.

**Figure 9 materials-15-02908-f009:**
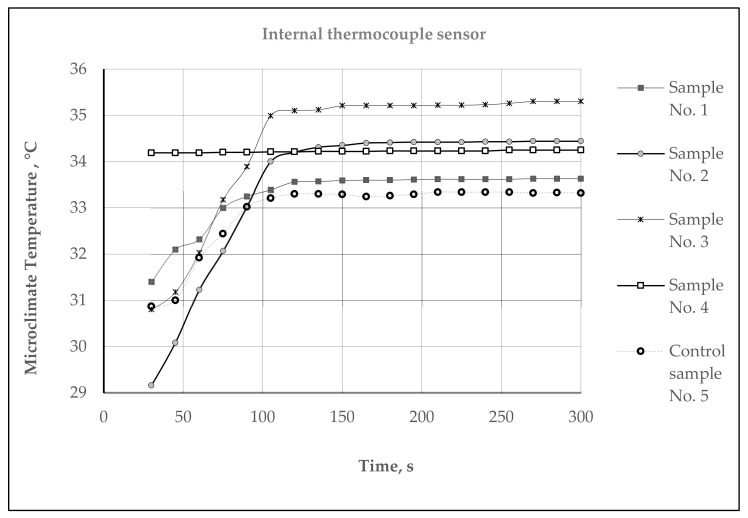
Dynamic curves of the microclimate temperatures of the different structures of socks.

**Figure 10 materials-15-02908-f010:**
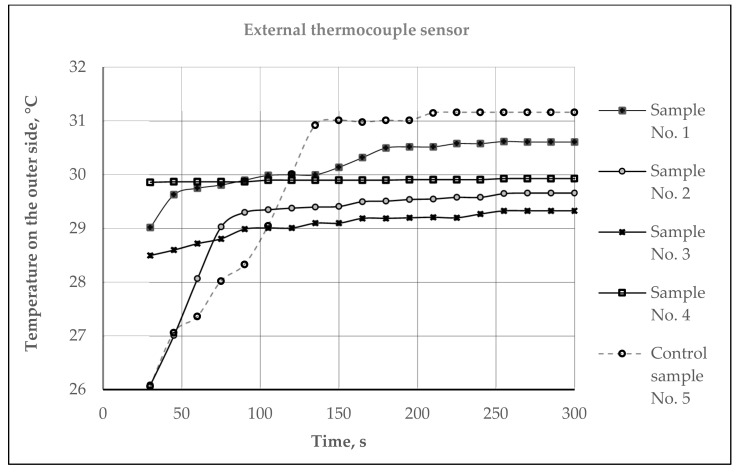
Dynamic curves of the heat exchange temperatures of different structures of socks.

**Figure 11 materials-15-02908-f011:**
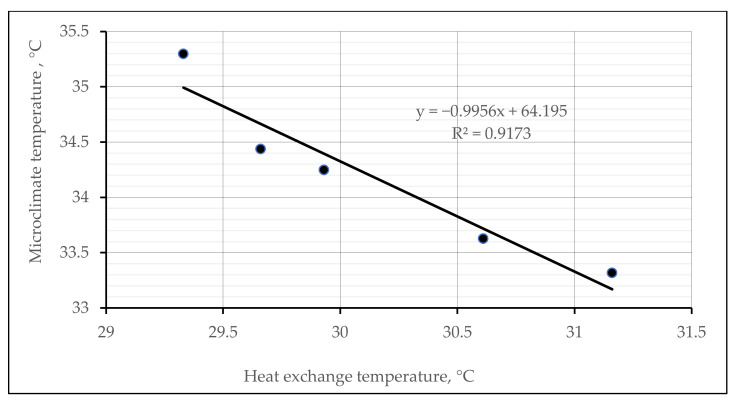
Relationship between the microclimate and heat exchange temperatures of different structures of socks.

**Figure 12 materials-15-02908-f012:**
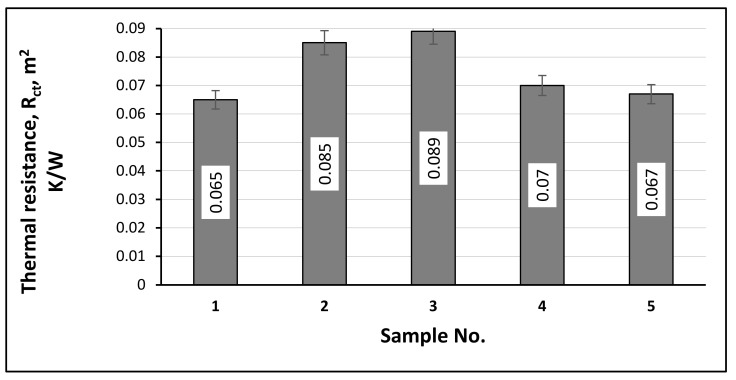
Thermal resistance of the different structures of socks.

**Figure 13 materials-15-02908-f013:**
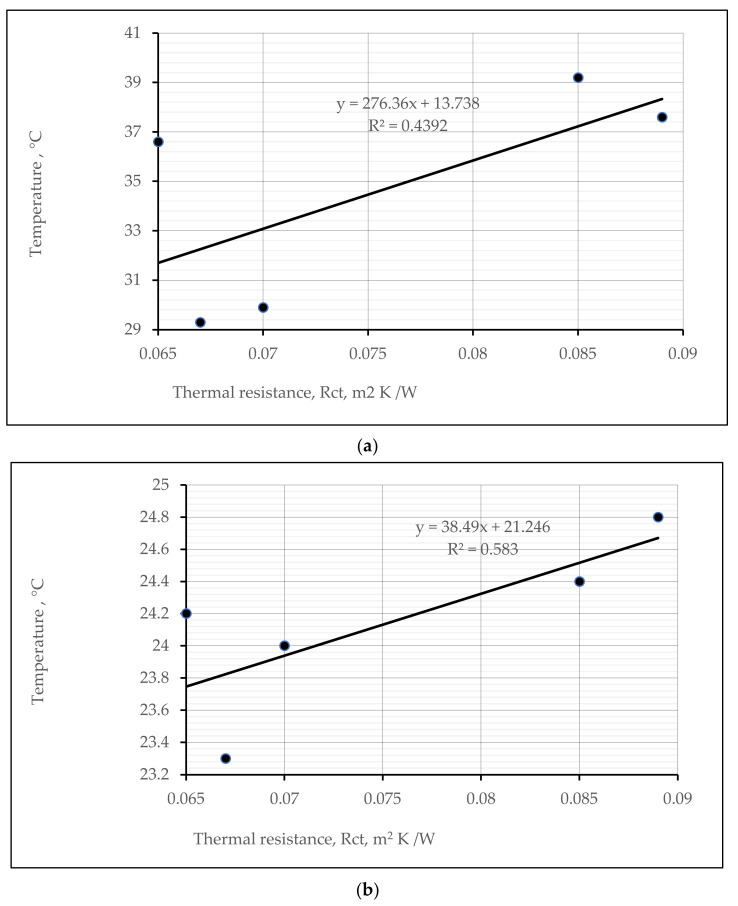
Relationships (by exposure to a 250 W IR lamp) between heat absorption at time 240 s temperature and thermal resistance (**a**), and heat retention at time 480 s for the temperature and thermal resistance (**b**) of different structures of socks.

**Figure 14 materials-15-02908-f014:**
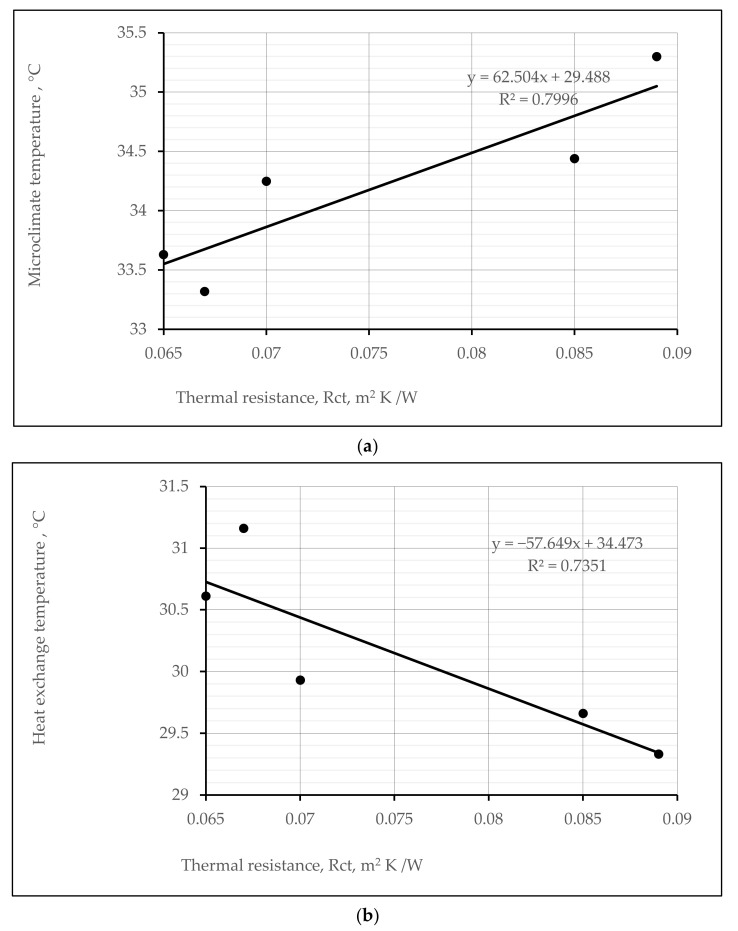
Relationships (at time 300 s) between the microclimate temperature and thermal resistance (**a**), and heat exchange temperature on the outer surface and thermal resistance (**b**) of different structures of socks.

**Figure 15 materials-15-02908-f015:**
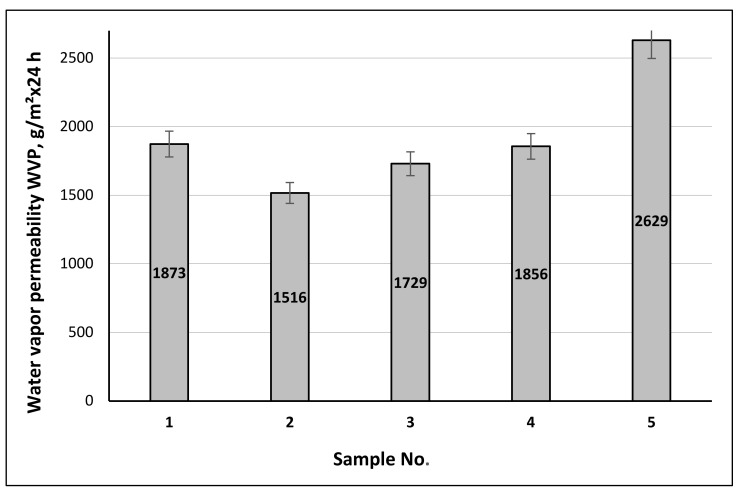
Water vapor permeability of various structures of the socks.

**Table 1 materials-15-02908-t001:** The main characteristics of the two-ply twisted yarns of the new structures.

No.	Composition of Two-Ply Twisted Yarns (S Direction; Twist Level—100 m^−1^): Type and Linear Density of Single Yarns	Total (Calculated) Linear Density, Tex
1.	(19.7 tex PES spun yarns Resistex^®^ Bioceramic + 17.0 tex twisted (filament count: 32 + 12), PES silver (Ag) coated multifilament yarns Shieldex^®^)	36.7
2.	(19.7 tex PES spun yarns Resistex^®^ Bioceramic + 7.8 tex (filament count—12) bicomponent PES/C, multifilament yarns with trilobal shaped carbon core Nega-Stat^®^ P190)	27.5
3.	(19.7 tex PES spun yarns Resistex^®^ Bioceramic + 11.2 tex (filament count—12) bicomponent PES/C multifilament yarns with trilobal carbon cross section, which 3 segments exits to PES fiber surface, Nega-Stat^®^ P210)	30.9
4.	(19.7 tex PES spun yarns Resistex^®^ Bioceramic + 19.7 tex PES spun yarns Resistex^®^ Bioceramic)	39.4
5.	For control sample: 11.1 tex textured PES yarns × 2	22.2

**Table 2 materials-15-02908-t002:** Structure characteristics of the main parts of the socks.

Plush Knitting Structure			Fiber Content in the Knitted Structure, %	Number of Stitches, cm^−1^		Mass per Unit Area, g/m^2^
Sample No.	Plush Yarn (Twist Direction S, Twist Level 100 m^−1^), Total Calculated Linear Density, R, Tex	Ground Yarn, Total Calculated Linear Density, R, Tex		Courses, P_v_	Wales, P_h_	
1.	(19.7 tex PES spun yarns Resistex^®^ Bioceramic + 17.0 tex twisted PES silver (Ag) coated multifilament yarns Shieldex^®^); R 36.7	(7.8 tex PA + 2.2 tex EL); R 10.0 yarns	Resistex^®^ Bioceramic—48 Shieldex^®^—42 PA—8 EL—2	11	10	480
2.	(19.7 tex PES spun yarns Resistex^®^ Bioceramic + 7.8 tex bicomponent PES/C multifilament yarns with trilobal shaped carbon core Nega-Stat^®^ P190); R 27.5	(7.8 tex PA + 2.2 tex EL); R 10.0 yarns	Resistex^®^ Bioceramic—62 Nega-Stat^®^ P190—25 PA—10 EL—3	10	10	365
3.	(19.7 tex PES spun yarns Resistex^®^ Bioceramic + 11.2 tex bicomponent PES/C multifilament yarns with trilobal carbon cross section, which 3 segments exits to PES fiber surface, Nega-Stat^®^ P210); R 30.9	(7.8 tex PA + 2.2 tex EL); R 10.0 yarns	Resistex^®^ Bioceramic—55 Nega-Stat^®^ P210—31 PA—11 EL—3	11	10	410
4.	(19.7 tex PES spun yarns Resistex^®^ Bioceramic + 19.7 tex PES spun yarns Resistex^®^ Bioceramic); R 39.4	(7.8 tex PA + 2.2 tex EL); R 10.0 yarns	Resistex^®^ Bioceramic—91 PA—7EL—2	9	10	460
5.	Control sample: 11.1 tex textured PES yarns × 2; R 22.2	(7.8 tex PA + 2.2 tex EL); R 10.0 yarns	PES—84 PA—12 EL—4	13	10	360

**Table 3 materials-15-02908-t003:** Electrostatic properties of socks of different structures.

Sample No.	Surface Resistivity, R, Ω	Specific Surface Resistivity, ƍ, Ω	Vertical Resistivity, R_v_, Ω	Shielding Factor, S	Half-Time Decay, t_50_, s
1	<2 × 10^3^	<2 × 10^3^	0.15 × 10^3^	1.00	<0.01
2	1.36 × 10^6^	2.69 × 10^7^	2.81 × 10^5^	0.00	>30
3	4.02 × 10^9^	7.96 × 10^10^	2.31 × 10^9^	0.96	<0.01
4	6.1 × 10^10^	12.08 × 10^11^	5.89 × 10^9^	0.00	1.62
5 (ontrol)	2.4 × 10^10^	4.75 × 10^11^	1.05 × 10^9^	0.00	0.59

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
