# Peer review of "Thermal Comfort and Electrostatic Properties of Socks Containing Fibers with Bio-Ceramic, Silver and Carbon Additives"

_materials, 2022, doi:10.3390/ma15082908_

Round 1

Reviewer 1 Report

This paper deals with the thermoregulation and antistatic properties of socks by using ceramic, silver and carbon structures. The present paper well-arranged and the results show a significant investigation.

However, some clarification can improve the current version of the manuscript.

1- Improve the conclusion part.

2- Please concise the introduction and put the novelty of the work at the end.

3- Brief description on figures and tables required.

4- Figures: the number of the figures should be decrease.

Figure captions should be revised based on the panels.

Figure 1: is not necessary. Can be deleted

Figure 2, 3:  can be in one Figure.

Figure 8, 9:  are confusing and the caption and the panel should be revised. In Figure 8: shows the irradiation part. In Figure 9: what is the third column for each sample. Please clarify all for the Figures.

5- Result and discussion: It should be revised based on the concept of the characterization and the presented Figures. Also, I feel that, although the authors have put a big effort in collecting a lot of results, there is a general lack of consistency and the content of the paper is often lost.

6- It can be helpful to investigate the hydrophobic/hydrophilic properties of the samples comparing to the permeability.

7- Which part of the socks that you mentioned in Figure 2 is crucial in thermoregulation and antistatic properties? Please clarify them.

8- The surface resistivity should be presented in Ω/sq. Please revise the Table 4 according to the correct units.

Reviewer 2 Report

This is an interesting paper describing the thermal and electro static properties of sock incorporated with bio ceramics, silver, and carbon. The abstract should be rewritten including the specific results of this study (what pattern, what concentration of additives, what thermal and electro static changes were observed, etc.). 

Further, it would have significantly added value to the experiments to include bacteria or fungus or odor experiments - all of which have easy assays - to determine the influence of these additives on socks. Durability of these additives could have also been addressed through repeating washing assays. SEM analysis of how the additives were incorporated into the socks would also help assess the results obtained. Lastly, all graphs should use statistical analysis methods.

Reviewer 3 Report

Comments to authors

In this manuscript, the authors investigated the thermal comfort and electrostatic properties of socks containing fibers with bio-ceramic, silver, and carbon additives. This manuscript is more like technical analysis to obtain the best composition or additives, rather than a scientific study. Bio-ceramic, sliver, and carbon, the functionalities of all these additives are well known. Thus, a major revision is highly recommended. Some comments were shown as follows:

1-Some figures were unclear and need to be redrawn, such as fig5 and fig6.

2-Standard deviation for the results needs to be provided. Otherwise, the data was unreliable.

3-For the conclusion, “All newly developed plush socks in their composition containing functional Resistex® Bioceramic, Nega-Stat® and Shieldex® fibers compared to the control sample is characterized by improved thermal efficiency” This is a predictable conclusion since it is obviously true if these materials are used. The authors should design the experiments to explain why the effect of the different positions of socks or different materials on the properties were different.

4-Many grammar errors have been found and need to be corrected.

Round 2

Reviewer 3 Report

Comments to authors

    There are still many grammar errors in the manuscript, which affects the understanding of the manuscript. Furthermore, the comparison with the previous studies in the discussion section needs to be added. A major revision was recommended. 

Author Response

Response to Reviewer 3 comments:

Point 1: There are still many grammar errors in the manuscript, which affects the understanding of the manuscript. Furthermore, the comparison with the previous studies in the discussion section needs to be added. A major revision was recommended. 

Response 1: We did our best by correcting English. Made some comparisons with previous studies.